

# Alaskan Stream flow in the eastern subarctic Pacific and the eastern Bering Sea and its impact on biological productivity

Sergey Prants, Andrey Andreev, Michael Uleysky, and Maxim Budyansky

Laboratory of Nonlinear Dynamical Systems, Pacific Oceanological Institute of the Russian Academy of Sciences, 43 Baltiyskaya st., 690041 Vladivostok, Russia, URL: http://dynalab.poi.dvo.ru

*Correspondence to:* S.V. Prants
(prants@poi.dvo.ru)

**Abstract.** We demonstrate the transport pathways of Alaskan Stream water in the eastern subarctic Pacific and the eastern Bering Sea from October 1, 1994 to September 12, 2016 with the help of altimetry-based Lagrangian maps. A mesoscale eddy activity along the shelf-deep basin boundaries in the Alaskan Stream region and the eastern Bering Sea is shown to be related with the wind stress curl in the northern North Pacific in winter. A significant correlation is found between the concentration of

chlorophyll *a* in the Alaskan Stream area and eastern Bering Sea in August – September and the wind stress curl in the northern North Pacific in November – March. The mesoscale dynamics, forced by the wind stress curl in winter, may determine not only lower-trophic-level organism biomass but also salmon abundance/catch in the study area.

## 1   Introduction

The Alaskan Stream (AS) is the northern boundary current of the North Pacific Subarctic Gyre flowing westward along the

shelf-break to the south of the Alaska Peninsula and the Aleutian archipelago. The AS is a narrow (<100 km), deep (>5000 m) and high-speed current with the transport of the order of $14$–$40 \cdot 10^6$ m$^3$ s$^{-1}$ (Favorite, 1974; Ueno et al., 2010). Portions of the AS flow through the Aleutian passes form the Aleutian North Slope Current (Fig. 1), the narrow and high speed current that flows northeastward along the north slope of the Aleutian Islands (Stabeno et al., 2009), and the Bering Slope Current flowing north-westward along the eastern shelf-break of the Bering Sea (BS) (Favorite, 1974; Stabeno and Reed, 1994; Johnson et al.,

2004). The northward flow of the AS water through the Aleutian passes is a main source of nutrients and heat for the BS ecosystem (Stabeno et al., 2005). The variations in the Alaska Gyre waters supply, caused by the AS, lead to interannual variations in the dissolved oxygen and temperature in the intermediate layer of the Okhotsk Sea and western subarctic Pacific area (Andreev and Baturina, 2006). Enhancement of the AS flow is accompanied by an increase in sea surface temperature and decreasing ice area in the Okhotsk Sea in winter and can be considered as direct and indirect causes of a reduction in the

chlorophyll *a* concentration and large-sized zooplankton biomass in the eastern Okhotsk Sea in winter-spring (Prants et al., 2015b).

Mesoscale variability is an important factor in the eastern subarctic Pacific and BS dynamics (e.g., Okkonen et al., 2001, 2003, 2004; Ladd et al., 2007). Mesoscale eddies enhance the cross-shelf exchange of macronutrients, iron and phytoplankton and zooplankton populations (e.g., Johnson et al., 2005; Okkonen et al., 2003). Crawford et al. (2005, 2007); Ueno et al.





(2010); Brown and Fiechter (2012) have indicated that such eddies play a significant role in controlling time and space patterns of chlorophyll *a* and may, therefore, determine the biological productivity and ecological function in the region. Interannual and decadal modulations of the eastern subarctic Pacific open-ocean ecosystems may be explained by analyzing statistics of eddy-induced cross-shelf transport (Combes et al., 2009). The impact of mesoscale eddies on the circulation and biology in the

eastern BS have been examined by many authors (e.g., Okkonen et al., 2004; Mizobata et al., 2002, 2006; Ladd et al., 2012). Instabilities in the Bering Slope Current, wind forcing, topographic interactions and flow through the eastern Aleutian passes have been suggested to be possible eddy-generation mechanisms. Analysis of the SSH time-series in 2002–2012 at the eastern boundary of the subarctic gyre demonstrated that the year-to-year changes of the SSH in the anticyclonic eddies were related to the wind stress curl in winter. It was assumed that spin up of the subarctic cyclonic gyre, forced by the wind stress curl, may

enhance the anticyclonic eddy activity in the AS area (Prants et al., 2013a).

In this study we focus at the AS transport pathways by using altimetry-based Lagrangian maps and forcing pattern that contributes to the interannual variability of the mesoscale dynamics and the chlorophyll *a* concentration in the east BS and AS area. We show that the intensity of AS anticyclonic eddies and anticyclonic eddies in the south-eastern BS is determined by the wind stress curl (WSC) in the northern North Pacific in November – March. There is a significant correlation between

the concentration of chlorophyll *a* at the deep basins margins in the BS and eastern subarctic Pacific in August – September and the WSC in the northern North Pacific in winter. Our results indicate that mesoscale dynamics in the eastern BS and AS areas may determine not only lower-trophic-level organism (the autotrophic phytoplankton) biomass but also the salmon abundance/catch.

The paper is organized as follows. Section 2 describes briefly the data we use and the Lagrangian methods we apply to study

transport pathways, origin, history and fate of different water masses. The next section 3 with the main results consists of two parts. Firstly, we study a correlation between the WSC in the northern North Pacific in winter and a mesoscale eddy activity along the deep basin boundaries in the AS region and the eastern BS. The altimetry-based Lagrangian simulation is used to track a penetration of AS and open-ocean waters into the eastern BS. Secondly, we study an impact of the mesoscale activity on chlorophyll *a* concentration in the area and its correlation with salmon abundance and catch. Section 4 discusses possible

physical mechanisms of the correlations found.

## 2 Data and methods

Geostrophic velocities and sea surface heights (SSHs) were obtained from the AVISO database (http://www.aviso.altimetry.fr) archived daily on a $1/4° \times 1/4°$ grid from October 1, 1994 to September 12, 2016. The distributed global product combines altimetric data from the TOPEX/POSEIDON mission, from Jason-1 for data after December 2001 and from Envisat for data

after March 2002. The meridional and zonal velocities and SSHs are gridded on a $1/4° \times 1/4°$ Mercator grid, with one data file every day.

To compute the WSC ($\mathrm{curl}_z(\partial\tau_y/\partial x - \partial\tau_x/\partial y)$, where $\tau_y$ and $\tau_x$ are respectively meridional and zonal wind stress components in the northern North Pacific) we used the monthly wind stress dataset from the NCEP reanalysis. Ocean chlorophyll *a*



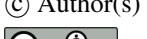

concentration data have been obtained from monthly composites from MODIS satellite imagery of ocean color (Level-3 product) with a horizontal resolution of 9 km on a regular grid (http://oceancolor.gsfc.nasa.gov). The salmon catch statistic was downloaded from the North Pacific Anadromous Fish Commission website (http://www.npafc.org). ARGO floats data (tracks,

seawater temperature and salinity) and bottle oceanographic data (temperature, salinity, nutrients and chlorophyll $a$ concentration) have been provided by the National Oceanographic Data Center (http://www.nodc.noaa.gov).

Lagrangian maps are geographic plots of Lagrangian indicators versus simulated particle's initial positions. We used before as Lagrangian indicators different functions of particle's trajectory. They have been shown in recent years (Prants, 2013; Prants et al., 2013b, 2017) to be useful to study large-scale transport and mixing in the ocean including propagation of radionuclides

in the western North Pacific after the accident at the Fukushima Nuclear Power Plant (Prants et al., 2011, 2014a; Prants, 2014; Budyansky et al., 2015) and finding of potential fishing grounds (Prants et al., 2014b, c). In order to track penetration of the AS and open-ocean waters into the eastern BS, we use in this paper a special kind of the Lagrangian maps which we call the origim maps. Being computed backward in time, the origin maps show where the waters came from to a study area and from which water masses a studied eddy consists of. They are computed as follows.

The vast area in the northern North Pacific, $50.0° \text{N} - 65.0° \text{N}$, $160.0° \text{E} - 145.0° \text{W}$, is seeded each three days with a large number of virtual particles for the period of time from October 1, 1994 to September 12, 2016. Their trajectories in the altimetric AVISO velocity field are computed backward in time for a year solving advection equations for passive particles with a fourth-order Runge – Kutta scheme

$$\frac{d\lambda}{dt} = u(\lambda, \varphi, t), \qquad \frac{d\varphi}{dt} = v(\lambda, \varphi, t), \tag{1}$$

where $u$ and $v$ are angular zonal and meridional altimetric geostrophic velocities, $\varphi$ and $\lambda$ are latitude and longitude, respectively. Bicubical spatial interpolation and third order Lagrangian polynomials in time are used to interpolate the velocity field.

We are interested in three water masses and their transport pathways. To track the AS waters, the section along the meridian $x_0 = 145° \text{W}$ from $y_0 = 58° \text{N}$ to $y_0 = 60° \text{N}$ is fixed. The particles, which crossed that section in the past, are colored in red

on the origin Lagrangian maps. The open-ocean particles, which crossed the section $x_0 = 160.0° \text{E} - 164.0° \text{W}$, $y_0 = 50.0° \text{N}$ in the past, are colored in green. The eastern BS particles, which crossed the section from $177.0° \text{E}$, $62.0° \text{N}$ to $164° \text{W}$, $55.0° \text{N}$ in the past, are colored in blue (see the yellow line Fig. 2a). We removed from consideration all the particles entered into any AVISO grid cell with two or more corners touching the land in order to avoid artifacts due to the inaccuracy of the altimetry-based velocity field near the coast. The corresponding colored Lagrangian maps demonstrate clearly origin, history and fate of

those water masses in the study area.

It is useful to identify locations in the ocean with zero geostrophic velocity. They are called in theory of dynamical systems as "elliptic and hyperbolic stagnation points" which are indicated on the Lagrangian maps by triangles and crosses, respectively. The elliptic points, located mainly in the centers of eddies, are those ones around which motion of water is stable and circular. The upward oriented blue triangles mark centers of anticyclones and downward oriented red mark cyclones. The hyperbolic points, located mainly between and around eddies, are unstable stagnation points with the direction along which water parcels





converge to such a point and another direction along which they diverge. The existence of hyperbolic areas in the ocean have
been recently confirmed by tracks of drifters in the western North Pacific (Prants et al., 2016).

## 3 Results

### 3.1 The Alaskan Stream eddies and transport pathways of the Alaskan Stream and open-ocean Pacific waters into the eastern Bering Sea revealed by the Lagrangian maps

The strong and narrow AS in the northern Pacific, the Aleutian North Slope Current and the Bering Slope Current in the BS are
clearly visible in the altimetric AVISO velocity field averaged for February (Fig. 1a) from 1994 to 2016. The main inflow of the
AS waters into the BS occurs through the Amchitka ($180°$ W) and Amukta ($172°$ W) Passes. In August, the surface circulation
in the eastern BS is determined by the mesoscale anticyclonic and cyclonic activity. The mesoscale anticyclones in the area
of the Bering Slope Current are topographically constrained with the Navarin, Zhemchug and Pribiloff canyons (Fig. 1b). An
intensification of the southwestward flow of AS water in the northern North Pacific and the north-northwestward flow of AS
water in the BS are typically observed in November – March when the Aleutian Low pressure cell is activated. Off-shore AS
meanders are related to the mesoscale anticyclonic and cyclonic eddies. The origin Lagrangian maps in Figs. 2a–c and 3a–d
and the surface salinity and nitrate distributions in Figs. 3e and f show that less saline and relatively low nitrate AS waters
(marked by the red color) intrude into the BS through the Aleutian passes and then flow northwestward along the Bering slope.
The small and large eddies in the Aleutian Islands area stimulate inflow of the AS water and the open-ocean subarctic water
(marked by green) into the BS. Due to impact of the coast, the Alaskan Coastal Current flow pathways in the Pacific Ocean
and BS (see, e.g., Schumacher et al., 1989; Stabeno et al., 2005) have not been analyzed by using altimetry-based Lagrangian
maps.

The southwestward drift of the AS mesoscale anticyclones along the Alaskan Peninsula and eastern Aleutian Islands forced
the AS water southward flow into the deep Pacific basin and enhanced the AS and open-ocean water to inflow in the BS (Figs. 1a
and b, Fig. 3a–d). We focused at the mesoscale anticyclonic eddies originated in the northern part of the Gulf of Alaska and
advected then by the AS along the ocean side of the eastern Aleutian Islands. These eddies have typically an elliptical shape
with a dimension of about $150 \times 200$ km with the centers (the elliptic points) located over the axis of the Aleutian bottom
trench. Inspecting the daily Lagrangian maps, we have found that one of such eddies, which we call ASAC 2003–2004, was
born in the northern part of the Gulf of Alaska in winter 2002. In summer and fall 2002, it was relatively weak and located
to the southwest off the Kodiak Island with the center at around $55.5°$ N, $153°$ W. The reinforcement and enlargement of this
eddy occurred in January – March 2003 southward of the Alaskan Peninsula in the area $53°$ N $- 55°$ N, $156°$ W $- 158°$ W when
the WSC over the northern North Pacific increased significantly.

The ASAC 2003–2004 is clearly seen on the origin Lagrangian map in Fig. 2a with the center at around $54.2°$ N, $157°$ W on
February 15, 2003. It consists of the core with a "white" water, a periphery with the AS "red" water surrounded by the "green"
open-ocean waters. The intensification of this eddy was accompanied by advection of the open-ocean water by a cyclonic eddy
toward the western and southern edges of the ASAC 2003–2004 (Figs. 2a and c) and thereby an increase of the temperature,



**Figure 1.** The altimetric AVISO velocity field averaged for a) February and b) August from 1994 to 2016. c) The Aleutian passes. Abbreviations: ACC — Alaska Coastal Current, ANSC — Aleutian North Slope Current, AS — Alaskan Stream, BSC — Bering Slope Current, N — Navarin Canyon, Z — Zhemchug Canyon and P — Pribilof Canyon.



salinity and density differences between the ASAC 2003–2004 and outside open-ocean waters (Figs. 2d–f). The ARGO CTD data (the buoys nos. 49070 and 4900176) demonstrate that the ASAC 2003–2004 had a warmer, less-saline and less-dense core

than the open-ocean subarctic water. During its southwestward drift in 2004, the ASAC 2003–2004 controlled a supply of the AS water (Fig. 2b) and open-ocean subarctic water into the BS through the Aleutian passes (Fig. 3b).

The eddy, which we call ASAC 2005–2006, was not advected to the $53°\,$N $- 55°\,$N, $156°\,$W $- 158°\,$W area in winter 2004 during the period of low WSC in the northern North Pacific but stayed southward of Kodiak Island and was relatively weak (Ladd et al., 2007). We observed its activation and reinforcement in winter 2005 when the WSC over the northern North Pacific

increased. During 2005 and 2006, the ASAC 2005–2006 forced the transport of AS and open-ocean waters into the BS (see Figs. 1Sa and b in Supplementary material). Figures 2Sa and b in Supplementary material show the vertical distributions of temperature, salinity and potential density in the ASAC 2005–2006 and outside waters in September 2005 and September 2006. Similar to the ASAC 2003–2004 (Figs. 2d–f), the ASAC 2005–2006 core was composed of relatively low salinity (33.7–33.9) and low density (26.7–26.9) waters. The temperature of waters inside of the anticyclone was $1 - 2$ degrees in Celsium higher

than outside it.

An increase of the WSC in winter 2008 led to enlargement and strengthening of the eddy ASAC 2008–2009 in the $53.5°\,$N $- 55°\,$N, $156°\,$W $- 158°\,$W region and formation of the second anticyclonic eddy. During its southwestward drift in 2008 and 2009, the anticyclonic eddies centered at $51.5°\,$N, $170°\,$W and $52.4°\,$N, $167°\,$W in August 16, 2009 forced the intrusions of AS water (summer 2008) and open-ocean Pacific water (summer 2009) into the BS (Figs. 3c and d). In winter 2010, the eastern anti-

cyclonic eddy was advected to the south off the Aleutian Islands and the western eddy drifted along the Aleutian Islands to the western subartic Pacific. In fall 2010 and winter 2011, the ASAC 2008–2009 has been observed in the central part of the Western Subarctic Gyre with the elliptic point at $51°\,$N, $170°\,$E.

The supply of the AS water through the Aleutian passes leads to formation and strengthening of anticyclonic eddies in the eastern BS. In January $-$ February 2003, an inflow of the AS water through the Amchitka Pass led to formation of the

anticyclonic eddy (the red patch centered at around $52.5°\,$N, $179°\,$W) in the southern BS (Fig. 2a). The generation of the "Pribiloff mesoscale anticyclonic eddy 2004" in the BS has been observed in the eastern Aleutian Passes in spring 2004. In May 2004 it intensified probably due to a density difference at its edges between the "red" AS water and "green" open-ocean water. In June 2004 its center was at the point $54.5°\,$N, $168°\,$W (see Fig. 1Sc in Supplementary material). In July $-$ September it occupied its position to the southwest of Pribiloff canyon with the elliptic points at $55.5°\,$N, $171°\,$W (Fig. 3b). In October 2004

it moved westward to the deep BS. During its staying in the Bering Slope Current region the Pribiloff mesoscale anticyclonic eddy 2004 trapped the "blue" BS shelf water and wound it around itself (Fig. 3b). The penetration of the BS outer shelf water into the eddy at that place has been shown by the CTD observations conducted in June 1997 (Ladd et al., 2012) (see Fig. 1Sd in Supplementary material).





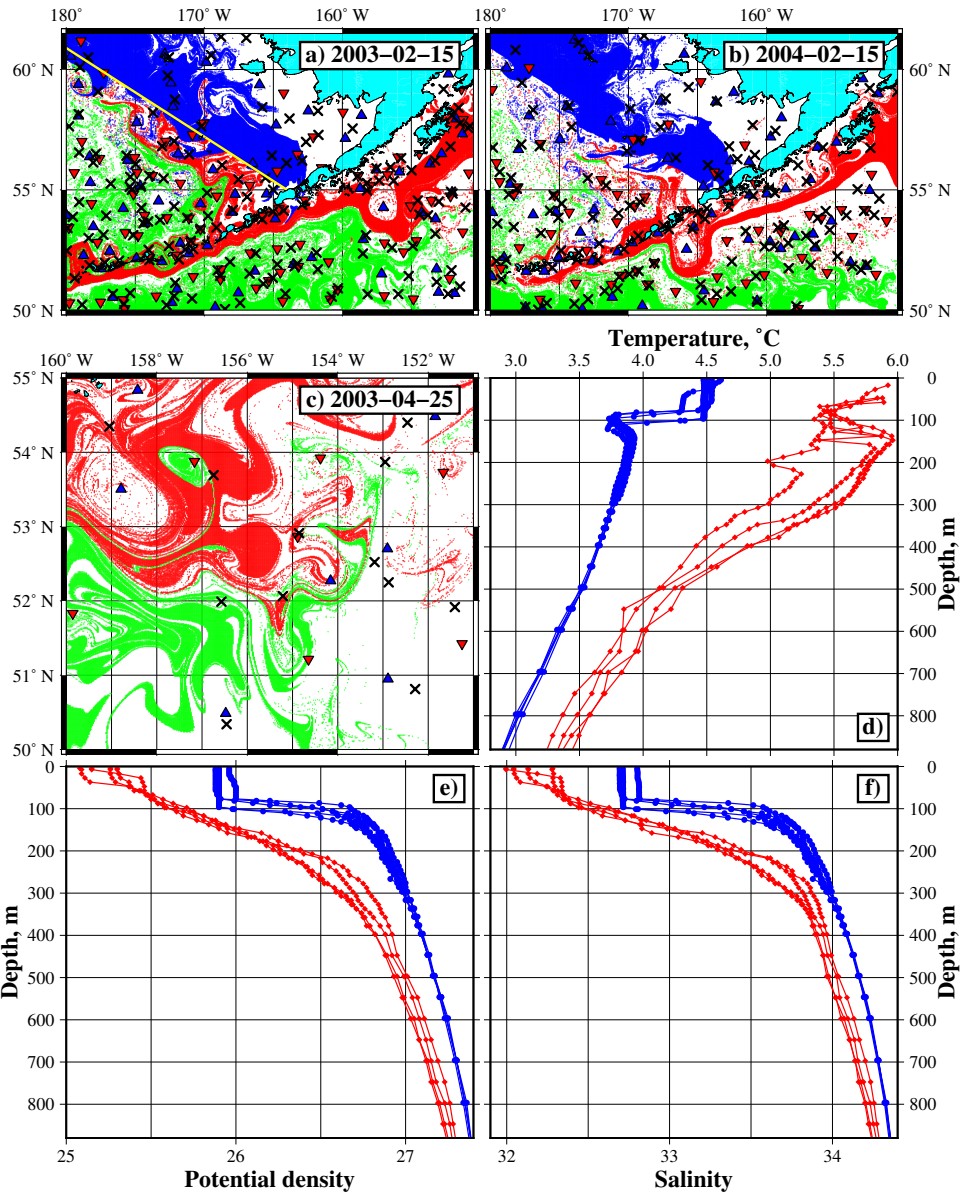

**Figure 2.** a) and b) The origin Lagrangian maps show the intrusion of AS waters (red) and off-shore subarctic waters (green) from the northern Pacific into the eastern BS through the Aleutian passes in February 2003 and February 2004. The penetration of the BS shelf waters into the deep basin of the eastern BS is demonstrated by the blue color. c) The map on April 24, 2003 with the AS anticyclone ASAC 2003–2004 centered at $53.5°$ N, $158.8°$ W and the eastern subarctic cyclone (the green patch of open-ocean water centered $51.2°$ N, $154.6°$ W). d)–f) The vertical distributions of temperature, salinity and relative density in the ASAC 2003–2004 (the red profiles) and the subarctic cyclone with open-ocean water (the blue profiles) in March – May 2003 (data from the ARGO buoys nos. 49070 and 4900176). Elliptic (stable) stagnation points with zero mean velocity at the fixed date are indicated by triangles.





**Figure 3.** a)–d) The origin Lagrangian maps demonstrate the impact of the AS anticyclonic eddies on the supply of "red" AS waters and "green" open-ocean subarctic waters from the northern Pacific into the eastern BS and penetration of the "blue" BS shelf waters into the deep BS basin in August and September 2003, 2004, 2008 and 2009. e) and f) The distributions of the salinity and nitrate concentrations in the surface waters (10 m) of the northern Alaska and eastern BS in summer (July – September).





### 3.2 A correlation between the mesoscale eddy activity in the Alaskan Stream region and the eastern Bering Sea and the wind stress curl in the northern North Pacific in winter

The strength and position of the Aleutian Low pressure cell are main factors which determine the circulation in the northern North Pacific. The strong Aleutian Low and positive WSC pattern over the northern North Pacific in November – March spin-up the subarctic cyclonic gyre (e.g., Ishi, 2005). One may assume that that spin-up results in more eddy activity at its northern
boundary (the AS region). The anticyclonic eddies contain a core of low-density water that produces an upward doming of the sea surface detectable by satellite altimeters.

The increased (decreased) SSH in AS anticyclonic eddies were associated with the increased (decreased) WSC in the northern North Pacific ($46° \, N - 48° \, N$, $165° \, E - 170° \, W$) in winter with the correlation coefficient $r = 0.75$–$0.90$, 2002–2016 (Fig. 4). Amplitude of the steric height variability in the study area is about 2.4 cm (the thermosteric height signal is about 2 cm and
halosteric height signal is about 0.4 cm) (Qiu, 2002) that is in several times less than the amplitude of the interannual variations of SSH in the AS area in February and August (around 20 cm) (Figs. 5a and b).

To find a correlation between the WSC and SSH and the velocities at the boundaries of AS anticyclonic eddies, we used monthly averaged SSH and velocities. If the correlation was found to be significant for two month period (May and June, July and August, etc.) we used the SSH and velocities averaged for May – June, July – August, etc. (see the corresponding figure
captions or legends). There is a good correlation with $r = 0.70$–$0.80$ between the monthly averaged SSH along the northern and northeastern boundaries of the Gulf of Alaska and monthly averaged zonal wind stress ($55° \, N - 60° \, N$, $140° \, W - 150° \, W$) in November – March. The formation of anticyclonic eddies along the northern boundary of the Gulf of Alaska can be related to the along-shore wind and downwelling (see, e.g., Combes and Di Lorenzo, 2007). However, our results show that reinforcement of the AS anticyclonic eddies southward of the Alaskan panhandle Peninsula ($53° \, N - 55° \, N$, $156° \, W - 158° \, W$) occurs during
the periods of increased WSC in the northern North Pacific. The correlation between the SSH in the AS anticyclonic eddies and the WSC is significant for two years (Fig. 4).

Our results demonstrate that the meridional and zonal velocities at the AS eddy boundaries during two years (while the eddies drift southwestward along the western Alaska Peninsula and eastern Aleutian Islands) are determined by the WSC in winter (Figs. 5c and d). The increased WSC in the northern North Pacific enhances the northward flow of the open-ocean subarctic water to the southern boundary of the AS anticyclonic eddies in January – February (Fig. 5e) and thereby increases the density gradient at the eddy edges.

The changes in the Aleutian Low activity and WSC in the northern North Pacific in winter determine year-to-year changes in velocities in some areas of the eastern BS. An increase (decrease) of the WSC in the North Pacific in November – March is accompanied by increased (decreased) velocities at the boundaries of the anticyclonic eddies in the central part of the deep BS in summer and fall (Fig. 3Sa). An intensification of the Aleutian Low and a large positive WSC result in increasing of the northward flow on the BS outer shelf in the areas located close to the Pribiloff, Zhemchug and Navarin canyons (Fig. 3Sc).
An increase (decrease) of the WSC in the northern North Pacific in November – March with a 1-year lag is accompanied by





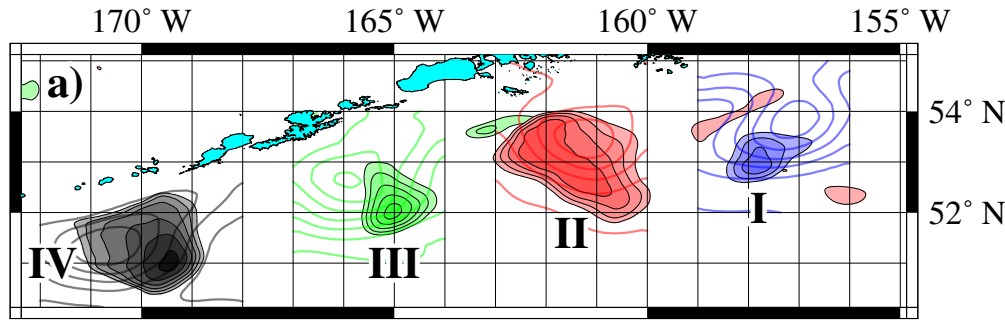

**Figure 4.** The distribution of the correlation coefficients (0.60–0.90, int. = 0.05) between the change of the WSC (November – March) and SSH in the AS area in March (I), August (II), February (III, 1-yr lagged WSC) and September (IV, 1-yr lagged WSC). The SSH isolines in March and August, 2003 and February and September, 2004.

increased (decreased) velocities at the boundaries of the anticyclonic eddies located in the area of Aleutian North Slope Current in summer and fall (Fig. 3Sb).

### 3.3 Chlorophyll *a* concentration and fish abundance and catch in the area

Mesoscale eddies modulate primary production along the eastern subarctic Pacific shelf break entraining chlorophyll *a*- and
iron-rich shelf water while simultaneously transporting nitrate- and silicate-rich basin water to the shelf (see, e.g., Okkonen et al., 2003; Ladd et al., 2005; Crawford et al., 2007). The dominant feature in chlorophyll *a* distribution in the surface layer in the AS region is a contrast between coastal and offshore waters. The coastal waters are productive with high values of chlorophyll *a* concentration (>3 µg/l in August and September), whereas the offshore waters are oligotrophic with low chlorophyll *a* concentration values (<1 µg/l).

The chlorophyll *a* pool southward of the Alaska Peninsula in summer 2005 and of the eastern Aleutian Islands in summer 2006 has been affected by the mesoscale anticyclonic eddies centered at $53.5°$ N, $161°$ W and $52.5°$ N, $167°$ W, respectively (Figs. 6a and b). The filaments with high chlorophyll *a* concentration (1–2 µg/l) have been transported off the shelf, wrapping around the mesoscale eddies and than being trapped inside the eddies. The WSC forcing in winter modulates the strength of anticyclonic eddies in the AS area and the velocities at their boundaries. Year-to-year variations in location and strength
of those anticyclonic eddies can determine spatio-temporal changes in chlorophyll *a* concentrations in the surface waters in the AS region during August – September. Figures 6c and d show year-to-year changes of the chlorophyll *a* concentration in August – September and the WSC in the northern North Pacific in November – March. Large (in 2003 and 2005) and small (in 2002 and 2004) values of the WSC in the northern North Pacific and, therefore, increased (decreased) velocities at the eddy's boundaries (Figs. 5c and d) have been accompanied by increased (decreased) chlorophyll *a* concentrations at the boundaries of
the AS eddies with no lag (Fig. 6c) and a 1-year lag (Fig. 6d).





**Figure 5.** a)–e) The year-to-year changes of the WSC (November – March) in the northern North Pacific, SSH, the meridional and zonal velocities in the AS area.





Figures 3Sa and b in Supplementary material show the distribution of chlorophyll $a$ concentration in the surface layer in the northern North Pacific and in the BS in August and September 2004 and a difference in the chlorophyll $a$ concentration between 2004 and 2003. Large values of the WSC over the northern North Pacific in winter 2003 resulted in strengthening of the AS anticyclonic eddy and change in the velocity field in the eastern BS in 2003 and 2004 while that eddy was drifting along the Aleutian Islands. In August and September 2004, the spots with high values (1.5–3 $\mu$g/l) of satellite-measured chlorophyll $a$ have been observed in the region of the Aleutian North Slope Current and along the shelf break of the eastern BS (Figs. 3Sa and b in Supplementary material). In August and September 2004, the surface waters have been significantly enriched by the chlorophyll $a$ pigment as compared to August and September 2003. In the central BS, we could see the shelf-break front

marking the boundary between low surface chlorophyll $a$ and relatively fresh outer shelf water and relatively high surface chlorophyll $a$ and more saline basin water. This front is biologically significant because it coincides with the BS "Green Belt", a region with high primary production that supports an extensive variety of consumer species (see, e.g., Springer et al., 1996; Okkonen et al., 2004)).

Year-to-year changes in the chlorophyll $a$ concentrations in the upper surface layer in the eastern BS (similar to the AS

region) have been positively correlated ($r = 0.7$–$0.8$, 2002–2016) with the WSC in the northern North Pacific in November – March (Fig. 6c). An intensification of the Aleutian Low in winter and thereby an increase of the WSC in the northern North Pacific were accompanied by increased chlorophyll $a$ concentration in the surface waters at the BS shelf edge with a 1.5-year lag).

Biomass of autotrophic plankton and concentration of chlorophyll $a$ are determined by many factors, such as solar radiation,

seawater temperature, macro- and micro-nutrients availability, water column stratification, etc. One of the most important factors, limiting phytoplankton growth in the upper layer in the subarctic North Pacific and the BS in the post-spring-bloom period (July – September), can be a low supply of nutrients with dissolved inorganic nitrogen and silicate considered to be the dominant elements limiting phytoplankton growth. In summer, there is a good agreement between the spatial distributions of salinity and the nitrate concentration in the surface layer in the BS and the eastern subarctic Pacific (Figs. 3e and f). Low salinity

(less than 32.2) Alaskan waters are associated with low nitrate concentrations (less than 4–6 $\mu$mol kg$^{-1}$) in the upper surface layer. In the subarctic North Pacific and the BS, the vertical distribution of salinity determines a density stratification in the upper layer. Existence of the strong vertical gradient of salinity (a halocline) limits vertical exchange between the surface and deeper layers. Due to tidal mixing in the Aleutian passes (such as the Seguam and Tanaga Passes) (Stabeno et al., 2005), salinity and the concentration of nitrate in the upper surface waters in the BS deep basin reach, respectively, 33 and 15–20 $\mu$mol kg$^{-1}$.

The nutrients, introduced due to mixing in the passes and then advected northward, are critical to the BS ecosystem.

The vertical profiles of the chlorophyll $a$ concentration in the eastern BS in summer are characterized by subsurface maximum located in the 10–40 m layer (Fig. 2Sd). The concentrations of chlorophyll $a$ in the upper 40 m layer reach 4–6 $\mu$g/l. The depth of the chlorophyll $a$ maximum location is related to nutrient availability and a water column stratification. The observations, conducted in July 2003 and 2004 in the eastern BS, demonstrate that the shallow (10–20 m) chlorophyll $a$ maxima have

been related to the waters with strong salinity and density stratifications and with salinities of 32.5–32.8 in the surface layer. For relatively low (32.3) and high surface salinities (33.0), the chlorophyll $a$ maxima have been located in the 20–40 m layer.



**Figure 6.** a) and b) Distributions of the chlorophyll *a* concentration (the MODIS data) imposed on the AVISO velocity field on the days indicated. c)–e) Year-to-year changes in the WSC (November – March) in the northern North Pacific and in the chlorophyll *a* concentration in the AS and BS area. Elliptic and hyperbolic (unstable) stagnation points with zero mean velocity are indicated by triangles and crosses, respectively.





The shallow location of the chlorophyll *a* maximum provides higher chlorophyll *a* concentration in the upper (0–10 m) surface layer captured by the satellite sensors.

Using the relationship between salinity and nitrate concentration (Ladd et al., 2005; Mordy et al., 2005; Stabeno et al., 2005), the difference in satellite-measured chlorophyll *a* concentration in the eastern BS between summer 2003 and summer 2004 (Figs. 3Sa and b in Supplementary material) can be explained. In summer 2003, the surface salinity was relatively low (32.4) in the most eastern part of the Bering deep basin ($54°$ N $– 55°$ N, $170°$ W $– 171°$ W) and relatively high (33.0) in its southern part ($53°$ N $– 54°$ N, $172°$ W $– 174°$ W) (Fig. 3Se). In summer 2004, the distribution of salinity in the surface layer in the eastern deep BS was quite uniform with the value of 32.8 (Fig. 3Se).

The salinity distributions in the deep eastern BS in summer 2003 and summer 2004 (Fig. 4Se) are in a good agreement with the water mass distributions demonstrated by the origin Lagrangian maps in Figs. 3a and b. The Lagrangian maps show that in summer 2003 the southeast of the BS was occupied by the "green" open-ocean subarctic (more saline) waters and populated mainly by cyclones, while the BSC area was occupied by the "red" AS (less saline) waters and populated by anticyclonic eddies (Fig. 3a). In summer 2004, the deep eastern BS was composed of spots of open-ocean and AS waters (Fig. 3a) and was characterized by a quite uniform salinity distribution possibly due to the anticyclones eddies activity (Fig. 4Se). The mixing, induced by anticyclonic eddies between the low salinity coastal water and high salinity deep basin water, probably created favourable conditions with nitrate availability and a shallow pycnocline for phytoplankton growth and, thereby, significantly increased the chlorophyll *a* concentration in the upper surface layer in the eastern BS in summer 2004.

The impact of the anticyclonic eddies on the chlorophyll *a* distribution in the AS area can be demonstrated by using a Lagrangian indicator $L = \int_0^T \sqrt{u^2 + v^2} dt$, which is a measure of a distance passed by advected particles. The indicator $L$ is more suitable for detecting and documenting vortex structures than the Lyapunov exponent and displacement of particles $D$ from their initial positions (Prants et al., 2015a, 2016). A studied area has been seeded at a fixed date with a large number of virtual particles whose trajectories have been computed backward in time for a month in the AVISO velocity field. The $L$ maps visualize not only the very vortex structures but also a history of water masses to be involved in the vortex motion in the past.

In Figures 7a and 5S in Supplementary material we impose distributions of the values of the Lagrangian indicator $L$ in the AS area on the chlorophyll *a* patterns in May 2006, May 2010 and May 2011 with an intermittency of productive coastal waters with high chlorophyll *a* values ($>6$ μg/l) and oligotrophic offshore waters with low chlorophyll *a* values ($<1$ μg/l). The black contours in those figures are isolines of $L$ with the step of 200 geographic minutes. They enclose stable mesoscale eddies, such as ones centered at $52.5°$ N, $165°$ W and at $53°$ N, $164°$ W. Filaments with high chlorophyll *a* concentration are transported offshore and wrapped around persistent mesoscale eddies existing in the area.

Mesoscale anticyclonic eddies with high primary production in the eastern subarctic Pacific and the eastern BS region are able to influence the zooplankton which could, in turn, support higher trophic levels and creates favorable fishing grounds. Distribution, migration paths and growth rate of salmon during its sea period of life are defined by oceanographic conditions at feeding grounds. Mesoscale activity (the scale and strength of eddies, intensity of mesoscale water transport, etc.) is one of the main factors which determines the dimension and spatial structure of salmon feeding grounds (see, e.g., Sobolevsky et al., 1994). Formation and dynamics of the salmon feeding base are influenced by the mesoscale activity in the region.





Eastern subarctic Pacific and the eastern BS are the main feeding areas for salmon in the northern North Pacific (see, e.g., Myers et al., 2007; Sato et al., 2009). Figures 7b–d demonstrate year-to-year changes in the WSC in November – March (the 5-years running mean) in the northern North Pacific and annual catch of chum salmon (b, d) and coho salmon (c). The total catch of chum salmon in the western Alaska area was comparatively low from about 1950 to 1970. Catches increased dramatically in 1975–1995 but declined in the late 1990s and slightly increased in 2005–2009 (Figs. 7b and d). The total catches of coho salmon in the central, southwestern and eastern Alaska region and of chum salmon in the western BS were relatively high

5 (30–50 thousands ton and 2–5 thousands ton) from 1980 to mid-1990s but significantly decreased to about 20 thousands ton and 0.5–1 thousands ton in 2000–2010 (Fig. 7c). Strong correlations between the changes in the chum and coho salmon catches in the eastern subarctic Pacific and the BS and the WSC in the northern North Pacific in winter with the coefficient $r = 0.64$–$0.75$ could be related to changes in the mesoscale eddy activity in the study area.

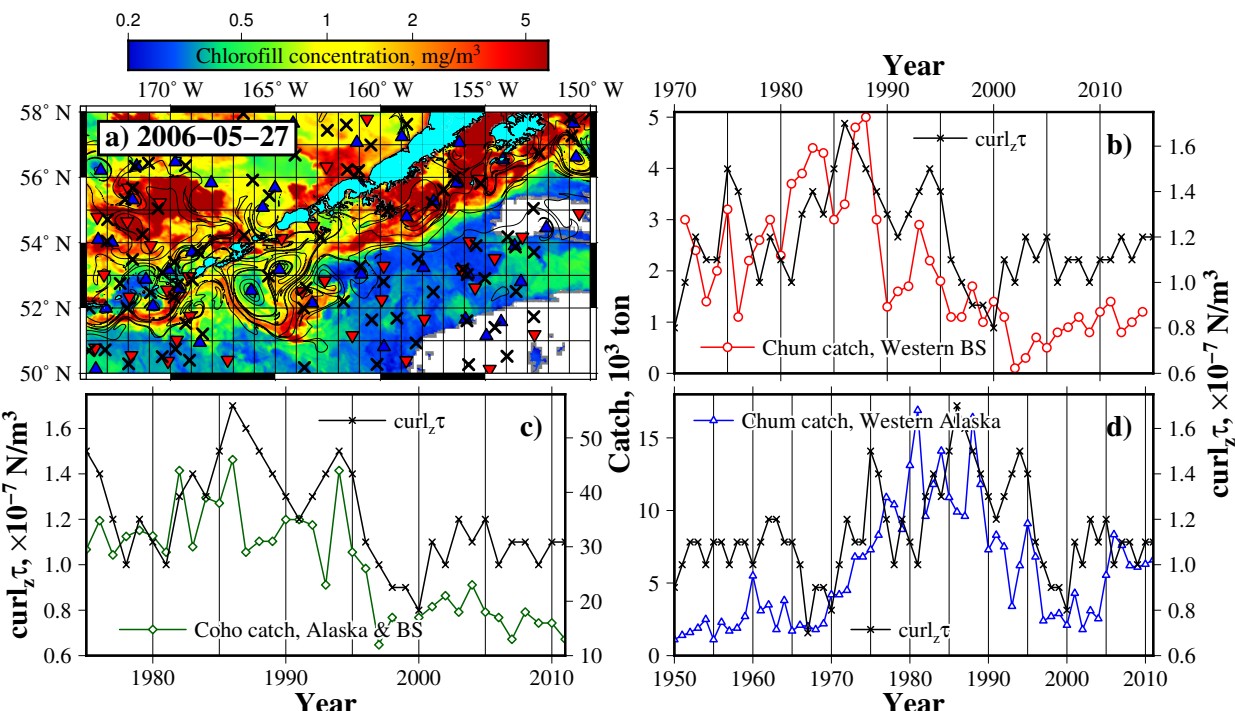

**Figure 7.** a) The impact of anticyclonic eddies on the chlorophyll *a* distribution (the MODIS data) in the AS area. The black contours are isolines of the Lagrangian indicator $L$ with the step of 200 geographic minutes. Year-to-year changes in the WSC in November – March (the 5-years running mean) in the northern North Pacific and in the annual catch of b) and d) chum salmon and c) coho salmon in the eastern subarctic Pacific and BS.





## 4 Discussion

The altimetry-based daily computed Lagrangian maps allow to track origin and transport pathways of the AS waters in the northern North Pacific and the BS and visualize mesoscale eddies in the study area. An intensification of the AS flow has been observed in November – March when the Aleutian Low was developed in the northern North Pacific, and strong positive WSC appeared in the subarctic North Pacific and the BS. In spring – fall, the southward meanders of the AS and the flow of the AS water and open-ocean subarctic water through the Aleutian passes into the BS have been caused by a mesoscale eddy activity.

Periods of increased eddy activity along margins of the eastern subarctic Pacific may be related to anomalous downwelling wind conditions along the continental margin (see, e.g., Combes and Di Lorenzo, 2007; Henson and Thomas, 2008). Increased poleward (downwelling favourable) wind stress, increased WSC along the eastern boundary and subsequent increase in the Alaska Current transport and intensification of the cyclonic gyre generate more mesoscale eddies (Okkonen et al., 2001; Melsom et al., 2003; Ladd et al., 2007). Wind stress curl is expected to be the most important forcing factor for the large-scale variability of circulation in the eastern subarctic Pacific (Cummins and Mysak, 1988). Combes et al. (2009) concluded that on interannual and longer time scales, the offshore transport of passive particles in the Alaskan Stream does not correlate neither with a large-scale atmospheric forcing, nor with local winds. In contrast, in the Alaska Current region, stronger offshore transport of passive particles coincides with periods of stronger downwelling which triggers development of stronger eddies.

Our results show that strength of the AS anticyclonic eddies (SSH in the eddy's center and velocities at the boundaries) to the south off the Alaskan Peninsula and the eastern Aleutian Islands are determined by the WSC in the northern North Pacific in November – March. One may assume that that spin-up of the cyclonic gyre in the subarctic Pacific, forced by the WSC in November – March, causes enhanced eddy activity along the continental slope of the Alaskan Peninsula and the Aleutian archipelago. The annual modulation in the cyclonic gyre's intensity in these areas should be interpreted as a barotropic response to the seasonal WSC forcing. The magnitude of this response is quantifiable by a time-dependent Sverdrup balance (Bond et al., 1994; Ishi, 2005). Reinforcement and strengthening of the AS eddies occur in the $53°$ N – $55°$ N, $156°$ W – $158°$ W area. The correlation between the SSH in the eddies, the velocities at their boundaries and the WSC in the northern North Pacific in winter is significant during two years while the eddies have been advected westward. An intensification of the anticyclonic eddies activity is accompanied by an increase of the northward advection of the open-ocean water to the eddy's boundaries and, thereby, an increase in the density difference between anticyclonic eddies and ambient waters.

The significant correlation between the surface velocities at the outer shelf margin of the eastern BS in summer and fall and the WSC in the North Pacific in winter (Fig. 3Sb) is related probably to the AS and open-ocean water supply to the BS. The increased inflow through the Aleutian passes may enhance eddy variability along the Bering Slope Current by increasing a baroclinic instability (Mizobata et al., 2008). The supply of low salinity Alaska coastal waters and relatively high salinity open-ocean waters creates zones with significant horizontal salinity and density gradients and, thereby, could enhance the mesoscale dynamics in the eastern BS.

An increase of the WSC in winter in the North Pacific activates anticyclone eddies in the central part of the deep BS in summer and fall (Fig. 3Sa) and (with a 1-year lag) the anticyclone eddies located in the Aleutian North Slope Current area



(Fig. 3Sc). Our results support the conclusion of Ladd et al. (2012) who indicated that the anticyclonic eddy activity along the eastern shelf-break of the BS (the Pribilof eddy) during the spring months is negatively correlated with the North Pacific Index, a measure of the strength of the Aleutian Low in November – March. Ladd et al. (2012) assumed that a spin-up of the subpolar gyre in the northern North Pacific leads to increased eddy activity in the BS, possibly due to a local effect of the stronger Bering

35   Slope Current or due to increased flow through the Aleutian passes.

Biological production in the deep basin of the BS is iron limited while the surface waters at the shelf are iron replete and nitrate limited (Aguilar-Islas et al., 2007). High surface chlorophyll *a* concentration along the shelf break in the BS appears to be associated with an eddy-induced mixing between shelf and deep basin waters (see, e.g., Okkonen et al., 2004; Mizobata et al., 2008; Ladd et al., 2012). The mixing, induced by anticyclonic eddies between the low salinity coastal water and high

salinity deep basin water, probably creates favourable conditions with a nitrate availability and a shallow pycnocline for the phytoplankton growth significantly increasing concentration of chlorophyll *a* in the upper surface layer in the eastern BS in summer (Figs. 3Sa, b and d). The canyons, located at the shelf break (the Bering, Pribilof, Zhemchuk and Navarin ones), are considered as preferred sites of cross-shelf exchange (Clement Kinney et al., 2009).

An increase of the WSC in the northern North Pacific in winter impacts the mesoscale dynamics in the eastern subarctic

Pacific and the eastern BS (Figs. 4 and 3S) and, thereby, a chlorophyll *a* concentration in surface waters (Fig. 6). Enhanced phytoplankton productivity in anticyclonic eddies may transfer up the food chain and results in increased biomass of higher trophic levels (zooplankton and fish). Advection paths of eggs and larvae can influence fish growth, survival and recruitment, either propelling them towards areas that support high growth and survival, or diverting them away from suitable habitat. For many fish species that spawn along the east subarctic continental slope, eggs and larvae benefit from slope to continental-shelf

transport, where larvae encounter favorable feeding and growth conditions prior to the onset of winter (Bailey et al., 2008; Atwood et al., 2010). General biological efficiency, defined by success of reproduction of organisms at the lowest trophic levels, has priority value for formation of steady salmon feeding conditions.

In the western part of the BS the highest biomass of chum salmon has been observed at periphery of anticyclonic eddies where zooplankton is accumulated in the upper 100-meter layer (Sobolevsky et al., 1994). Moss et al. (2013) demonstrated that

salmon, caught along the anticyclonic eddy's periphery in the eastern subarctic Pacific (the Sitka eddy), displayed the highest levels of insulin-like growth factor which is an index of the short-term growth rate for salmon. Zooplankton and phytoplankton densities are also greatest at the eddy's periphery. The location, timing and strength of the Sitka anticyclonic eddy, combined with juvenile salmon outmigration timing, could positively affect the growth by increased foraging opportunities. Years with enhanced production at the eddy's periphery and reduced inter- and intra-specific competition, resulting in increased survival

for certain stocks (Moss et al., 2013), are those when the three primary eddy features in the eastern subarctic Pacific (the Haida, Sitka and Yakutat eddies) have been located close to the shore during early summer months when juvenile salmon are migrating to the north. The observed correlations between the WSC and salmon catches for the eastern subarctic Pacific and the eastern BS area (Figs. 7b–d) may be explained by an intensification (slow down) of the mesoscale dynamics and cross shelf exchange by macro- and micro- nutrients, stimulating phytoplankton and zooplankton growth forced by increased (decreased) WSC in

the northern North Pacific in winter.



## 5    Conclusions

In this paper we proposed the forcing pattern that contributes to the interannual variability of the mesoscale dynamics and the chlorophyll *a* concentration in the surface waters in the Alaskan Stream area and the eastern Bering Sea. We conclude that the strength of the anticyclonic eddies along the deep basin slopes of the northern subarctic Pacific and the eastern Bering Sea is determined by the wind stress curl in the northern North Pacific in November – March. Strong correlations have been

found between the concentrations of chlorophyll *a* at the shelf – deep-sea boundaries of the Bering Sea and the northern Pacific subarctic in August – September and the wind stress curl in the northern North Pacific in November – March. Our results indicate that the mesoscale dynamics in the eastern subarctic Pacific and the eastern Bering Sea areas may determine not only lower-trophic-level organism (autotrophic phytoplankton) biomass but also salmon abundance and catch.

*Acknowledgements.*    This work was supported by the Russian Science Foundation (project no. 16–17–10025). The altimeter products were

distributed by AVISO with support from CNES.

Supplementary materials associated with this paper can be found in the on-line version.



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
