# Peer review of "Alaskan Stream flow in the eastern subarctic Pacific and the eastern Bering Sea and its impact on biological productivity"

_Biogeosciences, 2017_

## Referee Comment (RC1) · Anonymous Referee #1 · 7 Mar 2018

This manuscript seems problematic to me in terms of being appropriate to publish in Biogeosciences. It is primarily a physical oceanographic treatment of the relationship between wind stress curl and eddy formation, and how that affects the transport of waters through the key Aleutian passes into the Bering Sea. Chlorophyll data are used and an attempt is made to tie wind stress to oceanographic productivity in following seasons, but the chlorophyll data are largely limited to near surface waters by the satellite platform used. Some bottle data are used from oceanographic sampling, although the figures and text do not make it clear what the seasonal coverage is or the density of sampling is for the nutrient and chlorophyll data that were collected in this manner. Fisheries catch data tend to show that annual salmon landings were associated with

prior seasonal windier conditions, which makes a certain amount of sense, but there was no attempt to consider the several year lifespan of salmon in waters of the north Pacific and how that would affect lag time analyses, and the presentation of these linkages is not rigorous. For a paper in Biogeosciences, I think better connections need to be made with the large body of data that are available for nutrients, chlorophyll and fisheries practices in the north Pacific and Bering Sea.The manuscript is reasonably well written, but could benefit from some light Native English language editing. I provide some of those editing suggestions in my line comments below, although I did not edit the manuscript comprehensively.

Page 2, Line 11. Forcing pattern should be forcing patterns

Line 12. change contributes to contribute

Line 15, change basins to basin

Line 21, delete "a" before mesoscale

Line 23, change "a penetration" to the penetration; also change "an impact" to impacts

Page 3, Line 8, change "to study" to "for studying"

Lines 8-11. Given the widespread use of Lagrangian analyses in oceanography, citations should be to a review or limited citations, not just a number of citations to only the authors of this paper.

Line 13. "origim" should be origin

Line 24. Strictly speaking, this is about particle analysis rather than identification of three water masses.

Line 31-32, change "They are called in theory of dynamical systems as elliptic and hyperbolic stagnation points which" to "In the theory of dynamical systems, these can be termed elliptic and hyperbolic stagnation points that"

Line 33, change "are those ones around which motion of water is stable and circular" to "are those points around which the motion of water is stable and circular"

Line 34, change "downward oriented red mark cyclones" to downward oriented triangles in red indicate cyclones.

The next to final sentence of this section at the bottom of the page is not understandable.

Page 4, line 12 remove "the" before mesoscale anticyclonic and cyclonic activity.

Line 20, this is the first mention of the Alaska Coastal Current, which should be mentioned initially in the introduction.

Line 21, remove "by"

Line 24, remove "flow"

Line 26, change "We focused at the mesoscale anticyclonic eddies originated in the northern part of the Gulf of Alaska and advected then by the AS along the ocean side of the eastern Aleutian Islands" to "We focused on the mesoscale anticyclonic eddies originating in the northern part of the Gulf of Alaska and advected by the AS along the Pacific Ocean side of the eastern Aleutian Islands."

Line 28-29. Change "was born" to "originated"

Page 6, line 5. Delete "a" before supply

Page 6, line 4. Delete "the" before buoys

Line 14, change "in Celsium" to °C

Line 17, change "the second anticyclonic eddy" to a second anticyclonic eddy.

Line 20, change off to of

Line 26. Change "has been observed" to was observed. Also Pribilof has only one "f"

at its end

Line 30-31. Change "During its staying in the Bering Slope Current region the Pribiloff mesoscale anticyclonic eddy 2004 trapped the "blue" BS shelf water and wound it around itself (Fig. 3b)." to During its stay in the Bering Slope Current region, the Pribilof mesoscale anticyclonic eddy 2004 trapped the "blue" BS shelf water and wound around it (Fig. 3b)

Figure 2 caption. Add information on the x symbols following the description of the triangles and describe differences between the inverted and upwards pointing triangles. I know this is in the methods, but it would be helpful to repeat here.

Figure 3e, f. It is not clear to me what the sources of the salinity and nitrate data are, and this should be included in the caption. Also the density of the station locations should also be shown on the figures themselves, so the reader can evaluate the contouring.

Page 9, first line of text. Insert "the" before main factors

Line 7, change "were" to are

Line 8. In the text it is stated that the correlation coefficients range from 0.75 to 0.9, but on the figure, the stated range is 0.60 to 0.9.

Line 10, change "that is in" to ", which is"

Line 13, insert "a" between "for two"

Lines 15-17. How do we know this isn't just a casual correlation that really represents the predominant wind stress over the whole year and is not necessarily linked to the wind stress months earlier?

Line 19. Delete panhandle. Alaska Peninsula is sufficient. Alaska Panhandle usually refers to southeast Alaska, not southwest Alaska.

[Figure]

Line 29. Again Pribilof has only one f at the end.

Page 10, starting at line 20, change "The chlorophyll a pool southward of the Alaska Peninsula in summer 2005 and of the eastern Aleutian Islands in summer 2006 has been affected by the mesoscale anticyclonic eddies centered" to "The chlorophyll a pool south of the Alaska Peninsula in summer 2005 and south of the eastern Aleutian Islands in summer 2006 was affected by the mesoscale anticyclonic eddies centered…"

Page 12, line 26-27. The subsurface chlorophyll maximum is one of the key uncertainties in this paper, since according to the methods (and Figure 6), surface chlorophyll from the MODIS satellite sensors was used to determine spatial chlorophyll distributions, so the deep chlorophyll maximum was typically not observed, and it is challenging to know what can be validly said about chlorophyll concentrations over the whole euphotic zone relative to wind stress.

Page 14, first line makes reference to the chlorophyll maximum being shallower and leading to higher chlorophyll estimates from the MODIS sensors, but the immediately previous text at the end of page 12 doesn't seem to indicate that the chlorophyll maximum would have been recorded by the satellite sensors because it was too deep.

Top half of page 14. There is a relationship between nitrate and salinity, with more saline waters holding more nutrients, but it is also sensitive to biological utilization. Depending upon the degree of stratification, seasonal use of the nutrients and wind mixing, these are pretty speculative arguments to make, especially given the limited nature of the chlorophyll data available, primarily from satellite sensors.

Page 14, line 27. Creates should be create

Page 16, first line of discussion, change "The altimetry-based 10 daily computed Lagrangian maps allow to track origin and transport pathways of the AS waters in the northern North Pacific and the BS and visualize mesoscale eddies in the study area" to "The altimetry-based 10 daily computed Lagrangian maps allow tracking of the origin

and transport pathways of the AS waters in the northern North Pacific and the BS and facilitate visualization of mesoscale eddies in the study area."

Change next sentence: "An intensification of the AS flow has been observed in November – March when the Aleutian Low was developed in the northern North Pacific, and strong positive WSC appeared in the subarctic North Pacific and the BS." Change to: "An intensification of the AS flow was observed in November – March when the Aleutian Low developed in the northern North Pacific, and strong positive WSC appeared in the subarctic North Pacific and the BS."

Next sentence. Change "have been caused by a mesoscale eddy activity" to "were caused by mesoscale eddy activity"

Page 17, second sentence of second paragraph. Change concentration to concentrations

2nd paragraph, last sentence. Change "considered as preferred sites" to considered to be preferred sites.

Last paragraph of the discussion. This introduces discussion of other distant eddies such as those off Sitka and Yakutat, as well as observations from the western Bering Sea. This is poorly linked to the observations reported in the paper and need to be better integrated, if kept.
* * *

---

## Author Comment (AC1) · 29 Mar 2018

**The author's respond to the Reviewer #1**

We are very grateful to the Reviewer for a very careful reading of the manuscript and a number of useful corrections and suggestions.

**I.** *Reviewer comments: manuscript seems problematic to me in terms of being appropriate to publish in Biogeosciences. It is primarily a physical oceanographic treatment of the relationship between wind stress curl and eddy formation, and how that affects the transport of waters through the key Aleutian passes into the Bering Sea. Chlorophyll data are used and an attempt is made to tie wind stress to oceanographic productivity in following seasons, but the chlorophyll data are largely limited to near surface waters by the satellite platform used. Some bottle data are used from oceanographic sampling, although the figures and text do not make it clear what the seasonal coverage is or the density of sampling is for the nutrient and chlorophyll data that were collected in this manner. Fisheries catch data tend to show that annual salmon landings were associated with prior seasonal windier conditions, which makes a certain amount of sense, but there was no attempt to consider the several year lifespan of salmon in waters of the north Pacific and how that would affect lag time analyses, and the presentation of these linkages is not rigorous. For a paper in Biogeosciences, I think better connections need to be made with the large body of data that are available for nutrients, chlorophyll and fisheries practices in the north Pacific and Bering Sea.*

**Author's respond.**

Using altimetry-based Lagrangian maps and altimetric velocity field we show the Alaskan Stream, open-ocean and outer Bering Sea shelf water pathways in the subarctic Pacific and the Bering Sea. It's important and novel results for an area that has not been the focus of many studies before. In particular, we track the origin and composition of waters inside and outside of mesoscale eddies in the area. The mesoscale eddies are known to impact the spatial distributions of phyto- and zooplankton, fish, sea birds, and sea animals in the World Ocean. The subarctic North Pacific and Bering Sea are feeding and migration areas for Pacific salmon (pink, chum, sockeye, coho, and chinook). We present a driving force that leads to anticyclonic-eddy reinforcement and thereby to changes in lower-trophic-level phytoplankton and higher-trophic-level (salmon) organism biomass in the Alaskan Stream area and the eastern Bering Sea. It was not done before (it is cited from Combes et al., 2009: "on interannual and longer time scales, the offshore transport of passive tracers in the Alaskan Stream does not correlate neither with a large-scale atmospheric forcing, nor with local winds").

We do not relate salmon abundance/catches and windy conditions in the previous winter. We demonstrate the relationship between the salmon catches and the **5-yr running mean wind stress curl** in November-March (Fig. 7). The wind stress curl (Bond et al., 1994; Ishi, 2005) and windy conditions are different things. We show that the mesoscale eddy activity in the study area is due to the wind stress curl in November-March when the Aleutian Low develops in the northern North Pacific. By using **5-yr running mean** wind stress curl we indicate that an intensive mesoscale circulation and strong mesoscale eddies during a few years (not in the previous winter) possibly provide good conditions for salmon survival and growth. We refer to the results by Sobolevsky et al. (1994) and Moss et al. (2013). In the western part of the Bering Sea the highest biomass of chum salmon has been observed at periphery of anticyclonic eddies where zooplankton is accumulated in the upper 100-meter layer (Sobolevsky et al., 1994). Moss et al. (2013) demonstrated that salmon, caught at periphery of anticyclonic eddies in the eastern subarctic Pacific, displayed the highest level of the insulin-like growth factor which is an index of the short-term growth rate for salmon. Zooplankton and phytoplankton densities are also greatest at the eddy's periphery.

In our study we used the satellite and cruise chlorophyll data (WOD 2013). We indicated the data and coordinates of used chlorophyll data (Fig. 4S).

**II.** ***Reviewer comments***: *The manuscript is reasonably well written, but could benefit from some light Native English language editing. I provide some of those editing suggestions in my line comments below, although I did not edit the manuscript comprehensively.*
1) *Page 2, Line 11. Forcing pattern should be forcing patterns*
**Done**

2) *Line 12. change "contributes" to "contribute"*
**Done**

3) *Line 15, change "basins" to "basin"*
**Done**

4) *Line 21, delete "a" before "mesoscale"*
**Done**

5) *Line 23, change "a penetration" to "the penetration"; also change "an impact" to "impacts"*
**Done**

6) *Page 3, Line 8, change "to study" to "for studying"*
**Done**

7) *Lines 8-11. Given the widespread use of Lagrangian analyses in oceanography, citations should be to a review or limited citations, not just a number of citations to only the authors of this paper.*
**We reduced the number of references on *Lagrangian analyses.***

8) *Line 13. "origim" should be "origin"*
**Done**

9) *Line 24. Strictly speaking, this is about particle analysis rather than identification of three water masses.*
**We edited the text there accordingly to replace "the origin of water masses" by their transport pathways.**

10) *Line 31-32, change "They are called in theory of dynamical systems as elliptic and hyperbolic stagnation points which" to "In the theory of dynamical systems, these can be termed elliptic and hyperbolic stagnation points that"*
**Done**

11) *Line 33, change "are those ones around which motion of water is stable and circular" to "are those points around which the motion of water is stable and circular"*
**Done**

12) *Line 34, change "downward oriented red mark cyclones" to "downward oriented triangles in red indicate cyclones".*
**Done**

13) *The next to final sentence of this section at the bottom of the page is not understandable.*

[Figure]

**It is the illustration of organization of a flow around the hyperbolic point at (0,0) with the direction along which water parcels converge to such a point and another direction along which they diverge.**

14) *Page 4, line 12 remove "the" before "mesoscale anticyclonic and cyclonic activity".*
**Done**

15) *Line 20, this is the first mention of the Alaska Coastal Current, which should be mentioned initially in the introduction.*
**Done**

16) *Line 21, remove "by"*
**Done**

17) *Line 24, remove "flow"*
**Done**

18) *Line 26, change "We focused at the mesoscale anticyclonic eddies originated in the northern part of the Gulf of Alaska and advected then by the AS along the ocean side of the eastern Aleutian Islands" to "We focused on the mesoscale anticyclonic eddies originating in the northern part of the Gulf of Alaska and advected by the AS along the Pacific Ocean side of the eastern Aleutian Islands."*
**Done**

19) *Line 28-29. Change "was born" to "originated"*
**Done**

20) *Page 6, line 5. Delete "a" before "supply"*
**Done**

21) *Page 6, line 4. Delete "the" before "buoys"*
**Done**

22) *Line 14, change "in Celsium" to "_C"*
**Done**

23) *Line 17, change "the second anticyclonic eddy" to "a second anticyclonic eddy".*
**Done**

24) *Line 20, change "off" to "of"*
**Done**

25) *Line 26. Change "has been observed" to "was observed".*
*Also Pribilof has only one "f" at its end* (6 стр, 25 строка)
**Done**

26) *Line 30-31. Change "During its staying in the Bering Slope Current region the Pribiloff mesoscale anticyclonic eddy 2004 trapped the "blue" BS shelf water and wound it around itself (Fig. 3b)." to "During its stay in the Bering Slope Current region, the Pribilof mesoscale anticyclonic eddy 2004 trapped the "blue" BS shelf water and wound around it (Fig. 3b)"*
**Done**

27) *Figure 2 caption. Add information on the x symbols following the description of the triangles and describe differences between the inverted and upwards pointing triangles. I know this is in the methods, but it would be helpful to repeat here.*
**It was added to Figure 2 caption … Elliptic (stable) stagnation points with zero mean velocity at the fixed date are indicated by the downward and upward oriented triangles which mark cyclones and anticyclones, respectively.**

28) *Figure 3e, f. It is not clear to me what the sources of the salinity and nitrate data are, and this should be included in the caption. Also the density of the station locations should also be shown on the figures themselves, so the reader can evaluate the contouring.*
**It was added to Figure 3e, f captions…The distributions of the salinity and nitrate concentrations in the surface waters (10 m) of the northern Alaska and eastern BS in summer (July – September, World Ocean Atlas 2013, 1°× 1° grid).**

29) *Page 9, first line of text. Insert "the" before "main factors"*
**Done**

30) *Line 7, change "were" to "are"*
**Done**

31) *Line 8. In the text it is stated that the correlation coefficients range from 0.75 to 0.9, but on the figure, the stated range is 0.60 to 0.9.*
**In the text we corrected the correlation coefficients to be in the range from 0.6 to 0.9,**

32) *Line 10, change "that is in" to ", which is"*
**Done**

33) *Line 13, insert "a" between "for two"*
**Done**

*34) Lines 15-17. How do we know this isn't just a casual correlation that really represents the predominant wind stress over the whole year and is not necessarily linked to the wind stress months earlier?*

**There is a good correlation with r = 0.7 – 0.8 between the monthly averaged SSH along the northern and northeastern boundaries of the Gulf of Alaska and monthly averaged zonal wind stress (55N – 60 N, 140W – 150W) in November – March when the Aleutian Low developed in the northern North Pacific (Qiu, 2002).**

*35) Line 19. Delete panhandle. Alaska Peninsula is sufficient. Alaska Panhandle usually refers to southeast Alaska, not southwest Alaska.*
**Done.**

*36) Line 29. Again Pribilof has only one f at the end.*
**Done.**

*37) Page 10, starting at line 20, change "The chlorophyll a pool southward of the Alaska Peninsula in summer 2005 and of the eastern Aleutian Islands in summer 2006 has been affected by the mesoscale anticyclonic eddies centered" to "The chlorophyll a pool south of the Alaska Peninsula in summer 2005 and south of the eastern Aleutian Islands in summer 2006 was affected by the mesoscale anticyclonic eddies centered. . ."*
**Done.**

*38) Page 12, line 26-27. The subsurface chlorophyll maximum is one of the key uncertainties in this paper, since according to the methods (and Figure 6), surface chlorophyll from the MODIS satellite sensors was used to determine spatial chlorophyll distributions, so the deep chlorophyll maximum was typically not observed, and it is challenging to know what can be validly said about chlorophyll concentrations over the whole euphotic zone relative to wind stress.*

*Page 14, first line makes reference to the chlorophyll maximum being shallower and leading to higher chlorophyll estimates from the MODIS sensors, but the immediately previous text at the end of page 12 doesn't seem to indicate that the chlorophyll maximum would have been recorded by the satellite sensors because it was too deep.*
**We use the cruise chlorophyll data provided by the National Oceanographic Data Center (http://www.nodc.noaa.gov) (Data and Methods) when discussing the chlorophyll vertical profiles (Figure 4S). Page 12. "The vertical profiles of the chlorophyll a concentration in the eastern Bering Sea in summer are characterized by a subsurface maximum located in the 10–40 m layer (Fig. 2Sd)" was changed to "The vertical profiles of the chlorophyll a concentration in the eastern Bering Sea in summer are characterized by a subsurface maximum located in the 10–40 m layer (Fig. 4Sd)".**

*39) Top half of page 14. There is a relationship between nitrate and salinity, with more saline waters holding more nutrients, but it is also sensitive to biological utilization. Depending upon the degree of stratification, seasonal use of the nutrients and wind mixing, these are pretty speculative arguments to make, especially given the limited nature of the chlorophyll data available, primarily from satellite sensors.*
**We cannot agree with that by the following reasons:**
**a) We do not use the wind mixing. We use the wind stress curl which determines a spin up/down of the subarctic gyre (Bond et al., 1994; Ishi et al., 2005).**

**b) We demonstrate certain vertical distributions of chlorophyll using not only satellite data but the cruise data provided by the WOD2013 as well (see Fig. 4S). We show that higher satellite chlorophyll concentrations correspond more shallow location of the chlorophyll maximum (summer 2004 vs summer 2003). Moreover, we explain why there is a shallow chlorophyll maximum in the Bering Sea in summer 2004. In Data and Methods section we indicate that bottle oceanographic data (temperature, salinity, nutrients and chlorophyll a concentration) have been provided by the National Oceanographic Data Center (http://www.nodc.noaa.gov).**

**c) In the subarctic North Pacific and Bering Sea, vertical distribution of salinity is a proxy of density stratification (a temperature input to the density stratification is not significant). The relationship between salinity and nutrients (Andreev et al., 2002; Mordy et al., 2005; Ladd et al., 2012 etc.) reflects a balance between the export fluxes of organic matter from euphotic layer and the nutrient supply from deep to surface layers through the halocline (due to tidal and wind mixing, breaking of internal waves, instability of currents etc.). High salinity corresponds to a higher nutrient concentration and *vise versa* due to increased stratification and export fluxes in summer (e.g. Andreev A.G. et al. Vertical fluxes of nutrients and carbon through the halocline in the western subarctic Gyre calculated by mass balance. Deep-Sea Res. II. 2002. 49: 5577- 5593). We have not seen in the subarctic North Pacific any significant seasonal deviations from nitrate, silicate, phosphate and salinity relationship (Andreev et al., 2002).**

*40) Page 14, line 27. "Creates" should be "create"*
**Done**

*41) Page 16, first line of discussion, change "The altimetry-based daily computed Lagrangian maps allow to track origin and transport pathways of the AS waters in the northern North Pacific and the BS and visualize mesoscale eddies in the study area" to "The altimetry-based daily computed Lagrangian maps allow tracking of the origin and transport pathways of the AS waters in the northern North Pacific and the BS and facilitate visualization of mesoscale eddies in the study area."*
**Done.**

*42) Change next sentence: "An intensification of the AS flow has been observed in November– March when the Aleutian Low was developed in the northern North Pacific, and strong positive WSC appeared in the subarctic North Pacific and the BS." Change to: "An intensification of the AS flow was observed in November – March when the Aleutian Low developed in the northern North Pacific, and strong positive WSC appeared in the subarctic North Pacific and the BS."*
**Done.**

*43) Next sentence. Change "have been caused by a mesoscale eddy activity" to*
*"were caused by mesoscale eddy activity"*
**Done.**

*44) Page 17, second sentence of second paragraph. Change "concentration" to "concentrations"*
*2nd paragraph, last sentence. Change "considered as preferred sites" to*
*"considered to be preferred sites".*
**Done.**

45) *Last paragraph of the discussion. This introduces discussion of other distant eddies such as those off Sitka and Yakutat, as well as observations from the western Bering Sea. This is poorly linked to the observations reported in the paper and need to be better integrated, if kept.*

**We did not discuss the Sitka, Yakutat, and west Bering Sea eddies. We just referred the results by Sobolevsky et al. (1994) and Moss et al. (2013) in our discussion. Sobolevsky et al. (1994) demonstrated that the highest biomass of chum salmon has been observed in the Bering Sea at periphery of anticyclonic eddies where zooplankton is accumulated in the upper 100-meter layer. Moss et al. (2013) indicated that salmon, caught along the anticyclonic eddy's periphery in the eastern subarctic Pacific, displayed the highest level of the insulin-like growth factor which is an index of the short-term growth rate for salmon. Zooplankton and phytoplankton densities were also greatest at the eddy's periphery.**

---

## Editor Comment (EC1) · F. Wittmann (Editor) · 14 Jun 2018

Dear authors,

As I had real difficulties in finding additional reviewers for your manuscript I consulted the Chief Editors of Biogeosciences. Based on the comments of the reviewers of the former version and our own assessment we have jointly decided to reject your manuscript as out of scope for Biogeosciences. We think that your manuscript is primarily a physical oceanography study and recommend submitting it to a more specific journal, such as Ocean Science.